# Why Knowledge Distillation Works in Generative Models: A Minimal Working Explanation

**Sungmin Cha**[*]
New York University
sungmin.cha@nyu.edu

**Kyunghyun Cho**
New York University & Genentech
kyunghyun.cho@nyu.edu

## Abstract

Knowledge distillation (KD) is a core component in the training and deployment of modern generative models, particularly large language models (LLMs). While its empirical benefits are well documented—enabling smaller student models to emulate the performance of much larger teachers—the underlying mechanisms by which KD improves generative quality remain poorly understood. In this work, we present a minimal working explanation of KD in generative modeling. Using a controlled simulation with mixtures of Gaussians, we demonstrate that distillation induces a trade-off between precision and recall in the student model. As the teacher distribution becomes more selective, the student concentrates more probability mass on high-likelihood regions at the expense of coverage, which is a behavior modulated by a single entropy-controlling parameter. We then validate this effect in a large-scale language modeling setup using the SmolLM2 family of models. Empirical results reveal the same precision-recall dynamics observed in simulation, where precision corresponds to sample quality and recall to distributional coverage. This precision-recall trade-off in LLMs is found to be especially beneficial in scenarios where sample quality is more important than diversity, such as instruction tuning or downstream generation. Our analysis provides a simple and general explanation for the effectiveness of KD in generative modeling.

## 1 Introduction

Knowledge distillation (KD) has become a foundational technique in modern machine learning. Originally introduced as a method to compress large classification models by transferring knowledge from a teacher to a smaller student model [10], KD has since proven effective in improving generalization and aligning model behavior in several domains [21, 14, 23]. Its influence is especially prominent in generative modeling, where KD is now a standard component in the training and deployment of large language models (LLMs). From neural machine translation [12] to the latest instruction-tuned LLMs [13, 1, 8], distillation enables smaller models to generate coherent and high-quality text by mimicking the output distributions of larger models [9, 2, 26]. Despite its ubiquity, however, the mechanisms by which KD improves generative performance remain poorly understood, particularly in how distillation shapes the generative behavior of the student model.

While several studies have attempted to explain KD in the context of classification, emphasizing representation alignment, label smoothing effects, or decision boundary refinement [19, 17, 22], these analyses do not generalize naturally to autoregressive generative models. In particular, there is little theoretical understanding of how KD enables smaller generative models to achieve performance comparable to much larger models, even with significantly reduced capacity. Why do student models trained via KD generate higher-quality outputs than their maximum likelihood-trained counterparts? What inductive bias does the teacher introduce during distillation that improves sample quality?

---

[*]Code is available at: https://github.com/csm9493/kd-minimal-explanation

39th Conference on Neural Information Processing Systems (NeurIPS 2025).

These questions remain open, despite KD's critical role in the development of high-performing yet efficient LLMs.

In this paper, we provide a minimal working explanation of knowledge distillation in generative models by analyzing how distillation reshapes the student's learned distribution. We begin with a controlled simulation using Gaussian mixtures and show that distillation induces a trade-off between precision and recall: as the teacher distribution becomes lower in entropy, the student concentrates more probability mass on high-likelihood regions while sacrificing coverage. This behavior is governed by a single temperature-like parameter that controls the selectivity of the teacher's output. We then validate this insight in a large-scale language modeling setting. Specifically, we use SmolLM2 1.7B [3] as the ground-truth distribution to generate samples, on which we pretrain a 360M teacher model. This teacher is subsequently distilled into a 135M student model. Empirical results corroborate our theoretical predictions: as the teacher becomes more selective, the student produces sharper generations (*i.e.*, higher precision) at the cost of reduced recall—mirroring the trade-off observed in the Gaussian mixture framework. These findings suggest that distillation enables smaller generative models to concentrate probability mass on high-density regions of the output space, effectively producing sharper and more fluent generations. This trade-off is especially desirable in practical scenarios such as instruction tuning or task-specific generation, where sample quality is prioritized over full coverage. Our contributions are summarized as follows:

- We provide a minimal working explanation of knowledge distillation in generative models, highlighting a precision-recall trade-off that emerges from teacher entropy.
- Through a controlled Gaussian mixture setup, we show that distillation shifts the student's focus toward a selective subset of the data distribution emphasized by the teacher, leading to improved precision at the cost of recall.
- We validate this mechanism in large-scale language models, where precision-recall dynamics observed in simulation are replicated via multistage distillation from SmolLM2 1.7B to 360M to 135M.

## 2 Related Work

**Knowledge distillation.** Knowledge distillation (KD) was originally introduced as a model compression technique that transfers knowledge from a large teacher model into a smaller student [10]. By matching the teacher's softened output probabilities, the student can capture richer inter-class similarity patterns, leading to improved generalization and efficiency [4, 24]. This foundational idea has inspired a range of extensions. Born-Again Networks [7] demonstrate that students can outperform their teachers through repeated distillation, while FitNets [21] leverage intermediate feature representations to guide deeper students during training. Recent work further suggests that KD can improve generalization even when the teacher and student have identical capacities [11, 22], indicating that the utility of KD extends well beyond model compression.

**Broader use of KD across domains.** KD has become a central component in modern NLP systems and generative modeling. In neural machine translation, sequence-level KD was proposed to improve generation quality while reducing model size [12]. More recently, state-of-the-art LLM frameworks such as DISTILLM [13], Phi-4-Mini [1], and LLaMA 3 [8] have adopted KD as a core technique for post-training alignment and efficient deployment. Beyond language models, KD has also been applied in diverse domains such as self-supervised learning [23], speech recognition [6], and continual learning [14, 5], often to transfer behaviors from larger or prior models into smaller or adaptive ones. Despite its broad adoption, most implementations of KD treat it as a black-box heuristic, lacking a clear understanding of its internal mechanisms, particularly in generative settings.

**Why does KD work?** A number of theoretical efforts have attempted to explain why KD works. In linear models, soft targets have been shown to produce more robust and smoother decision boundaries [19]. KD has also been connected to label smoothing, suggesting that it introduces an implicit regularization effect [17]. Empirical studies have confirmed that KD improves generalization across a variety of conditions, even when teacher and student architectures are identical [22]. Additional work frames KD as a trade-off between knowledge inheritance and exploration [11]. However, these interpretations have been largely developed in the context of classification tasks. There remains a significant gap in understanding how KD influences the generative behavior of models, particularly autoregressive language models—a gap that our work aims to address.

Recent studies on KD for generative models have explored related concepts like mode-seeking behavior [9, 2, 26]. Often, achieving such behavior involved proposing specialized objectives, like reverse KL divergence [9, 26] or tailored loss functions [2]. In contrast, our work provides a different perspective by demonstrating how this precision-enhancing effect naturally arises within the *standard forward KL divergence* framework, simply by controlling the teacher distribution's selectivity. Furthermore, we analyze this phenomenon not just between teacher and student, but within a broader three-stage framework (ground truth $\rightarrow$ teacher $\rightarrow$ student), offering a fundamental, distribution-level explanation based on a precision-recall trade-off for why common KD practices improve sample quality relative to the original data distribution.

## 3 Distillation in Generative Modeling: Analysis via Gaussian Mixtures

To understand the core effect of knowledge distillation in generative modeling, we introduce a simple yet expressive setup based on mixtures of Gaussians. This construction allows us to precisely control the complexity of the data distribution and quantify how the student model responds to different teacher behaviors. By modulating the entropy of the teacher distribution through a temperature-like parameter $\beta$, we adjust how selectively the teacher emphasizes certain regions of the data space. This minimal setting reveals a key trade-off between precision and recall in the student model, providing insight into how distillation alters the student's learned distribution.

### 3.1 The precision-recall trade-off induced by distillation

Let $D = \{x_1, \ldots, x_N\}$ be a dataset sampled from a ground-truth distribution $p^*(x; \theta^*)$ defined as a mixture of $K$ Gaussians:

$$p^*(x; \theta^*) = \sum_{k=1}^{K} \alpha_k \, \mathcal{N}(x; \mu_k, \Sigma_k),$$

where $\alpha_k > 0$ for all $k$, and $\sum_{k=1}^{K} \alpha_k = 1$. We assume that the component means $\{\mu_k\}$ are distinct and that all covariance matrices $\Sigma_k$ are finite. Such a mixture model serves as a universal approximator for continuous distributions when $K$ is sufficiently large.

**Fitting a teacher model.** Assume we have fit a teacher distribution $p'(x; \theta')$ as a mixture of $K' \leq K$ Gaussians:

$$p'(x; \theta') = \sum_{k=1}^{K'} \alpha'_k \, \mathcal{N}(x; \mu'_k, \Sigma'_k), \tag{1}$$

using KL divergence minimization $\mathrm{KL}(p^* \| p')$. In the ideal case, there exists a (possibly one-to-many) mapping $\sigma : \{1, \ldots, K'\} \rightarrow \{1, \ldots, K\}^+$ such that each component $k'$ in the teacher distribution $p'$ covers a subset of components in $p^*$.

In the extreme case of $K' = K$, $\sigma$ would ideally be a permutation operator, and all the parameters are well recovered up to some *noise* due to the finite number of samples within $D$. This *noise* is an important aspect here, which implies that even if $\alpha_k = \frac{1}{K}$, we would never end up with the uniform $\alpha'_k$. In the other extreme case of $K' = 1$, $\sigma(k') = 1$ for all $k'$, and we end up with

$$\mu'_k = \frac{1}{N} \sum_{n=1}^{N} x_n, \quad \Sigma'_k = \frac{1}{N} \sum_{n=1}^{N} (x_n - \mu'_k)(x_n - \mu'_k)^{\top}.$$

An interesting aspect of this low-end extreme is that the region of high probability density concentration under $p'(x; \theta')$ with $K' = 1$ has minimal overlap with those of $p^*$. In other words, the samples drawn from $p^*$ will be lowly scored by $p'(x; \theta')$ with $K' = 1$ and vice-versa.

**A lower-entropic reparametrization of the teacher model.** To control the selectivity of the teacher distribution, we reparameterize the mixture weights $\alpha'_k$ using a temperature-like parameter $\beta \geq 1$:

$$\alpha'_k(\beta) = \frac{\exp(\beta \log \alpha'_k)}{\sum_{j=1}^{K'} \exp(\beta \log \alpha'_j)}.$$

When $\beta = 1$, this recovers the original mixture weights. As $\beta$ increases, the distribution over components becomes increasingly peaked, reducing entropy and concentrating mass on components with higher original weight. In the limit $\beta \to \infty$, this reduces to a deterministic selection of the highest-weight component:

$$\alpha'_k(\infty) = \begin{cases} 1, & \text{if } k = \arg\max_j \alpha'_j \\ 0, & \text{otherwise.} \end{cases}$$

Note that noise in learning (*e.g.* due to the finite sample $D$) implies that we would always end up with such an extreme case of zero-entropy $\alpha'_k(\infty)$. Even if we magically end up with a uniform $\alpha'_k$s, we can always add a small amount of noise to break the tie. As a result, this yields a modified teacher distribution,

$$p'(x; \theta', \beta) = \sum_{k=1}^{K'} \alpha'_k(\beta) \, \mathcal{N}(x; \mu'_k, \Sigma'_k),$$

which emphasizes a narrower subset of components as $\beta$ increases. In effect, larger $\beta$ values induce a lower-entropy teacher $p'(x; \theta', \beta \to \infty)$ that selectively focuses on a few modes of the original distribution, such as $\sigma(\arg\max_{k'} \alpha'_{k'})$. Samples from such a teacher are likely to lie in high-density regions under $p^*$ but do not reflect the full diversity of the underlying distribution. This reparameterization provides a simple knob to control the difficulty of the student's task: high-$\beta$ teachers provide cleaner, more concentrated training signals, while low-$\beta$ teachers preserve broader coverage. We refer to such entropy-controlled distributions as *teacher models* in the remainder of the paper.

**Training a student model.** Let us consider training a *student model* $p''(x; \theta'')$, which is also a mixture of $K'' \ll K$ Gaussians, against the teacher model by minimizing

$$\mathrm{KL}(p'(x; \beta) \| p''(x)) = - \underbrace{\int p'(x; \beta) \log p''(x) \mathrm{d}x}_{(a)} + \text{const.}, \tag{2}$$

where we omitted $\theta'$, since there is no confusion here. We will also omit $\theta''$ unless it is explicitly needed. To better understand this objective, we expand term (a) using Jensen's inequality:

$$\int p'(x; \beta) \log p''(x) \mathrm{d}x = \int \left( \sum_{k'=1}^{K'} \alpha'_{k'}(\beta) \mathcal{N}(x; \mu'_{k'}, \Sigma'_{k'}) \right) \log \left( \sum_{k''=1}^{K''} \alpha''_{k''} \mathcal{N}(x; \mu''_{k''}, \Sigma''_{k''}) \right) \mathrm{d}x$$

$$\geq \sum_{k'=1}^{K'} \sum_{k''=1}^{K''} \underbrace{\alpha'_{k'}(\beta) \alpha''_{k''}}_{(a')} \underbrace{\int \mathcal{N}(x; \mu'_{k'}, \Sigma'_{k'}) \log \mathcal{N}(x; \mu''_{k''}, \Sigma''_{k''}) \mathrm{d}x}_{(b')}.$$

Here, (a') captures the joint weighting of each pair of teacher and student components, while (b') measures the cross-entropy between individual Gaussians. Term (a') encourages alignment between the mixture coefficients of the student and teacher: components with larger $\alpha'_{k'}(\beta)$ exert more influence, pushing the student to allocate higher weights to the corresponding $\alpha''_{k''}$. Due to the normalization constraint $\sum_{k''} \alpha''_{k''} = 1$, student components not aligned with high-weight teacher modes are implicitly suppressed.

Term (b') incentivizes geometric alignment: each student component is encouraged to match the shape and location of teacher components it overlaps with. However, this matching is modulated by the weighting in (a'). Only those component pairs for which both $\alpha'_{k'}(\beta)$ and $\alpha''_{k''}$ are substantial will contribute meaningfully to the loss. Intuitively, this means that distillation focuses the student's capacity on faithfully representing the most emphasized regions of the teacher distribution.

This formulation reveals how the entropy of the teacher, controlled by $\beta$, governs the selectivity of the student's focus. Larger $\beta$ values lead to a more peaked teacher distribution, which in turn biases the student toward modeling fewer but sharper modes.

**Controlling the difficulty for training the student model.** As shown above, the influence of each teacher component on the student is governed by its corresponding mixture weight $\alpha'_{k'}(\beta)$.

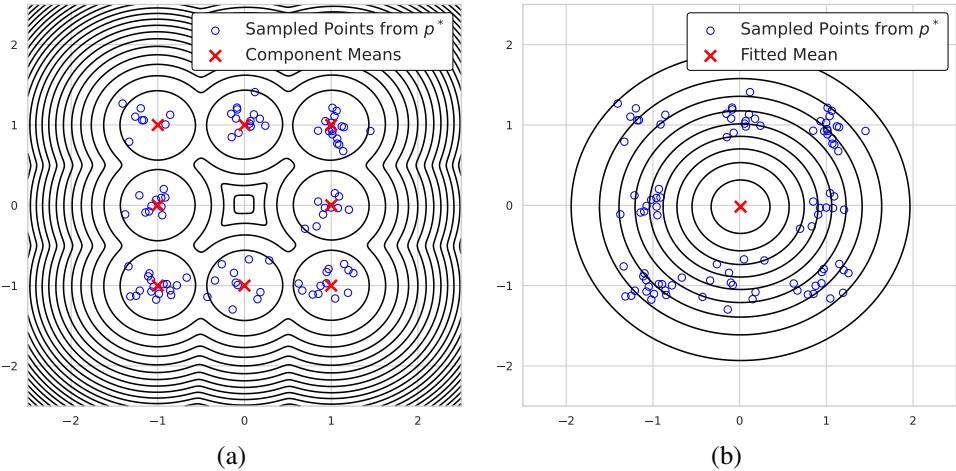

Figure 1: (a) Contours show the probability density of the ground-truth distribution $p^*$, with dots representing samples drawn from it. (b) The contours correspond to the student model trained directly without distillation.

When $\alpha'_{k'}(\beta)$ is close to zero, the student is effectively relieved from modeling the $k'$-th component, regardless of its support in the original distribution $p^*$. In other words, components of $p^*$ mapped to low-weight regions of the teacher via $\sigma(k')$ are unlikely to be covered by the student. Consequently, the student model is ignoring all the components of the original distribution $p^*$ included in $\sigma(k')$ (that is, the ones that were mapped to the $k'$-th component in the teacher model). The student model only needs to capture and cover those teacher's components with large $\alpha'_{k'}$'s, and thereby $\cup_{\alpha'_{k'} \geq 1-\epsilon}\sigma(k')$ of the original distribution $p^*$.

We define the difficulty of training the student model as the discrepancy between its capacity $K''$ (the number of student's components) and $\left|\cup_{\alpha'_{k'} \geq 1-\epsilon}\sigma(k')\right|$ (the number of active components in $p^*$ that are emphasized by the teacher):

$$\text{Difficulty} \propto K'' - \left|\bigcup_{\alpha'_{k'} \geq 1-\epsilon}\sigma(k')\right|.$$

This quantity closely correlates with the number of emphasized teacher components, *i.e.*, $\sum_{k'=1}^{K'}\mathbb{1}(\alpha'_{k'} \geq 1 - \epsilon)$. Crucially, this difficulty is directly modulated by the temperature-like parameter $\beta$, which controls the entropy of the teacher distribution. Higher $\beta$ values yield more selective (lower entropy) teachers, concentrating training signals on a smaller subset of $p^*$ and thereby reducing the student's modeling burden.

**The resulting student model.** Let $p''(x; \theta'', \beta)$ denote the student model trained to match the $\beta$-modulated teacher $p'(x; \theta', \beta)$. We evaluate the quality of the learned distribution using two complementary metrics:

$$\text{Precision}(\beta) = \mathbb{E}_{p''(x;\theta'',\beta)}\left[\log p^*(x;\theta^*)\right], \tag{3}$$

$$\text{Recall}(\beta) = \mathbb{E}_{p^*(x;\theta^*)}\left[\log p''(x;\theta'',\beta)\right]. \tag{4}$$

Intuitively, precision measures how well samples generated by the student are supported under the true distribution $p^*$, while recall quantifies how thoroughly the student covers the modes of $p^*$. As $\beta \to \infty$, the teacher distribution becomes increasingly selective, concentrating mass on a small subset of high-density regions. The student, in turn, learns to model these regions accurately—leading to high precision but reduced recall. Conversely, when $\beta \approx 1$, the teacher approximates the full support of $p^*$, and the student is encouraged to match the entire distribution. This maximizes recall but often comes at the cost of lower precision due to limited capacity.

By modulating $\beta$, distillation provides a simple mechanism to control this trade-off. Our analysis reveals that this precision-recall dynamic emerges naturally from the structure of the distillation

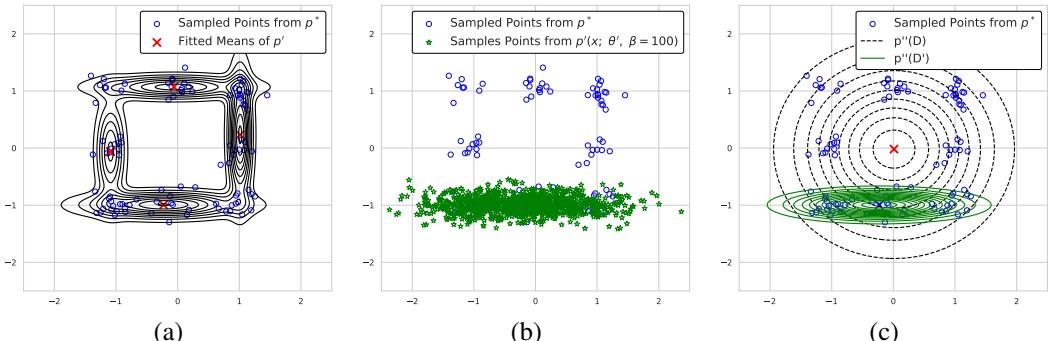

Figure 2: (a) Contour plot of the teacher distribution $p'$, with samples from the true distribution $p^*$ overlaid as blue dots. (b) Samples (green) drawn from the $\beta$-modulated teacher distribution $p'(x; \theta', \beta = 100)$, showing strong concentration on the bottom three modes of $p^*$. (c) Contours of the student models: the dashed black contour corresponds to a student trained directly on $p^*$ samples, while the green contour represents a student trained on teacher samples (distillation). The distilled student clearly focuses on a narrower region emphasized by the teacher. This setup corresponds to a low-difficulty case: with $\beta = 100$, the teacher concentrates on a single dominant component ($\alpha'_k \approx 1$), and the student has just enough capacity ($K'' = 1$) to match it, resulting in the Difficulty measure close to 0.

objective and offers a principled explanation for why distillation can improve sample quality in generative models, even under constrained capacity.

## 3.2 Simulation

To empirically illustrate the mechanisms discussed above, we construct a toy generative task using a mixture of Gaussians. The ground-truth distribution $p^*$ consists of eight well-separated components arranged in a rectangular grid (Fig. 1(a)). We begin by fitting a student model composed of a single Gaussian directly on samples from $p^*$. As shown in Fig. 1(b), this model places its density around the center of the space, an area with near-zero mass under $p^*$, demonstrating the difficulty of approximating a complex multimodal distribution with a simple model under standard MLE training.

Next, we fit a teacher model $p'$ with four Gaussian components. Each teacher component approximately covers two modes of $p^*$. The learned mixture weights are:

$$\alpha' = [0.15, 0.26, 0.24, 0.33].$$

We then sample training data from a temperature-modulated version of the teacher, $p'(x; \beta = 100)$. At this high value of $\beta$, sampling concentrates on the teacher component with the largest weight, yielding data that primarily covers the bottom three modes of $p^*$ (Fig. 2(a) and (b)).

A new student model is then trained on these samples. Fig. 2(c) compares the density contours of the directly trained student (dashed black) and the distilled student (green). The distilled model clearly focuses on a smaller region of $p^*$ where the teacher emphasized high-density modes, while ignoring other parts of the support.

To quantify this trade-off, we report the following metrics:

- **Direct student**: Precision = $-20.26$, Recall = $-2.64$

- **Distilled student** ($\beta = 100$): Precision = $-0.70$, Recall = $-42.45$

These results confirm our theoretical prediction: as the teacher becomes more selective (higher $\beta$), the student learns to place greater mass on high-likelihood regions at the cost of covering the full distribution. In conclusion, knowledge distillation enables simpler models to generate sharper, more concentrated outputs—a desirable property when sample quality is prioritized over full coverage.

# 4   Connection to Autoregressive Language Models

## 4.1   From Gaussian mixtures to autoregressive language models

While the previous section focused on synthetic data from Gaussian mixtures, the core mechanism we observed—namely, a trade-off between precision and recall modulated by teacher entropy—extends naturally to autoregressive language models. Distillation has long been a standard technique in training such models [12], and is now integral to large-scale language model (LLM) pipelines, from instruction tuning to efficient deployment in resource-constrained environments [13, 1, 8].

Our Gaussian mixture setup provides a useful abstraction: each component corresponds to a mode of the data distribution. In language models, this analogy holds at the token level, where the next-token distribution is modeled as a categorical distribution over a vocabulary. Formally, an autoregressive language model defines a joint distribution as:

$$p(x_1, x_2, \ldots, x_T) = \prod_{t=1}^{T} p(x_t | x_{<t}),$$

and can be reinterpreted as a mixture of trajectory conditionals:

$$p(x_2, \ldots, x_T) = \sum_{v \in V} p(x_1 = v) \cdot p(x_2, \ldots, x_T | x_1 = v),$$

mirroring the mixture structure seen in Gaussian models. Moreover, it is common practice to apply entropy-reducing techniques, such as temperature scaling, top-$k$, or top-$p$ sampling [25], to produce sharper generations. This is analogous to our $\beta$-modulated teacher distributions.

Prior work [27] further shows that the expressivity of each token distribution is upper-bounded by the dimensionality of the model's hidden states. Since this dimensionality scales with model size, smaller models inherently represent fewer modes, and thus face similar capacity bottlenecks to those seen in our mixture of Gaussian setup.

In the following section, we test whether the precision-recall trade-off observed in simulation also emerges in LLMs. We perform multistage distillation using the SmolLM2 [3] family of models, and examine how increasing teacher selectivity (controlled via a $\beta$-like parameter) affects the student's precision and recall.

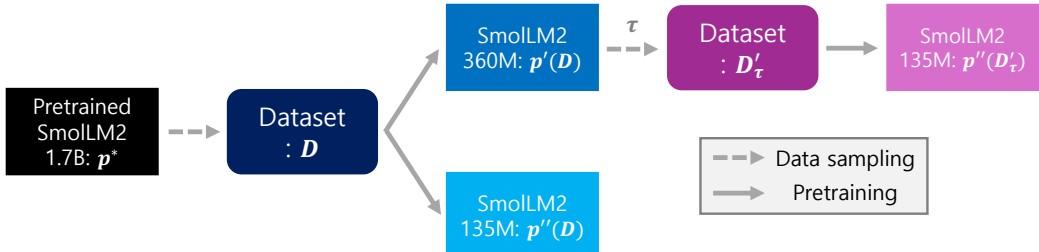

Figure 3:   Overview of our LLM distillation setup. We first treat the pretrained SmolLM2 1.7B model as the ground-truth distribution $p^*$ and sample 10M sequences to construct dataset $D$. We then pretrain a 360M teacher model $p'$ on $D$ using next-token prediction loss. To control the teacher's entropy, we sample from $p'$ with varying temperature values $\tau$ to generate distillation datasets $D'_\tau$. Finally, we train a 135M student model $p''$ on each $D'_\tau$ and evaluate its precision and recall with respect to $p^*$.

## 4.2   Experimental setup

To test whether the precision-recall dynamics observed in our synthetic setup also manifest in real-world models, we conduct a series of distillation experiments using the SmolLM2 [3] family. Fig. 3 provides an overview of our experimental pipeline, which mirrors the generative structure outlined in Section 3. We conduct each experiment with five different random seeds.

We begin by treating the pretrained SmolLM2 1.7B model as the ground-truth distribution $p^*$. To generate samples from $p^*$, we use the fixed prompt `"The"` as a consistent start-of-sequence token and

decode with temperature $\tau = 1.0$ and `top-k` set to the full vocabulary size. For each generation, we sample up to 512 tokens, yielding well-formed sentences. We repeat this process to generate 10M sequences, which we denote as the training dataset $D$ sampled from $p^*$.

To construct the teacher model $p'$ (Equation 1), we randomly initialize a 360M parameter SmolLM2 model and pretrain it from scratch on $D$ using the standard next-token prediction loss. Training is conducted using 4 GPUs with a total batch size of 256, optimized using AdamW [15] (initial learning rate 5e-4) with a warmup-square-decay scheduler [28]. We train for 5 epochs and follow hyperparameter settings from the original SmolLM2 release, with minor adjustments for training stability as detailed in the Supplementary. We select the best-performing teacher checkpoint based on the lowest perplexity evaluated on a held-out validation set $D_{\text{val}}$ of 100,000 sentences sampled independently from $p^*$.

Next, we distill this teacher model into a 135M student model $p''$ (in Equation 2) using temperature-scaled generations from $p'$. Specifically, we sample from the pretrained teacher $p'$ using multiple temperatures ($\tau \in \{0.8, 0.875, 0.95, 1.0\}$), which correspond to varying levels of entropy in the output distribution—lower temperatures produce more selective (*i.e.*, lower-entropy) teachers. For each temperature $\tau$, we generate a new dataset $D'_\tau$ consisting of 10M sequences. We then train the student model $p''$ from scratch on each $D'_\tau$ using the same training setup and hyperparameters as for the teacher.

Finally, we measure the student model's *precision* and *recall* with respect to the original 1.7B model $p^*$ using the definitions in Equations 3 and 4. Precision is computed using 100,000 validation sequences ($D''_{\text{val},\tau}$) sampled with `top-k` using temperature $\tau = 1.0$ from the student model $p''(D'_\tau)$ , while recall is measured using the same number of samples from the ground-truth $p^*$ (*i.e.*, from $D_{\text{val}}$). This evaluation allows us to observe how the precision-recall trade-off evolves as the teacher becomes more selective via temperature scaling.

### 4.3  Experimental result

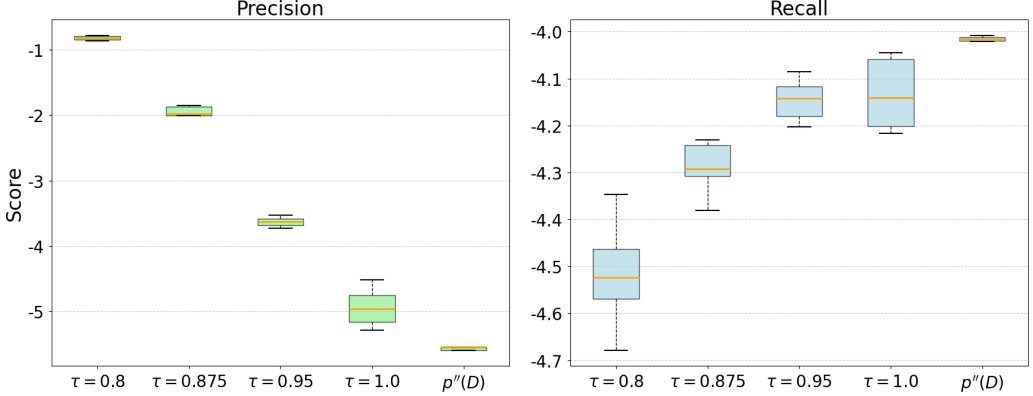

Figure 4:  Score distribution of Precision (left) and Recall (right) based on the $\tau$ parameter and the $P''(D)$ model. Each box plot illustrates the interquartile range across five seeds, with the orange line indicating the arithmetic mean. Higher (less negative) values on the y-axis denote better result.

Fig. 4 reports the precision and recall of student models $p''$ (SmolLM2 135M) distilled from the teacher $p'$ (SmolLM2 360M) under varying sampling temperatures $\tau$, with evaluation performed against the ground-truth model $p^*$ (pretrained SmolLM2 1.7B). Each $\tau$ in x-axis corresponds to a specific distillation setup, where $p''$ is trained on samples drawn from $p'$ using temperature $\tau$ ($p''(D'_\tau)$). The final one, $p''(D)$, represents a baseline where the student is trained directly on data sampled from $p^*$, without any intermediate teacher.

We observe that as the sampling temperature $\tau$ decreases, the student model $p''$ exhibits higher precision but lower recall, favoring high-likelihood regions while sacrificing coverage. This behavior suggests that lower temperatures lead the teacher to produce more peaked outputs, concentrating mass on dense modes, which the student mimics, thereby trading off coverage for sharpness. The result closely mirrors our earlier findings in the Gaussian mixture framework, where distillation from

a low-entropy teacher led the student to replicate only the most prominent modes of the original distribution.

These results provide empirical evidence that the geometric behavior of knowledge distillation—trading off recall for increased precision—persists even in large-scale language models. Rather than merely compressing the teacher's output space, distillation reshapes the student's distribution to emphasize its high-density core. This behavior is especially useful in scenarios where sample quality is more critical than diversity, such as instruction tuning, reasoning tasks, or downstream generation tasks. Our findings thus reinforce the interpretation of distillation as a mechanism for inducing controlled distributional concentration, consistent with the Gaussian mixture framework analyzed earlier.

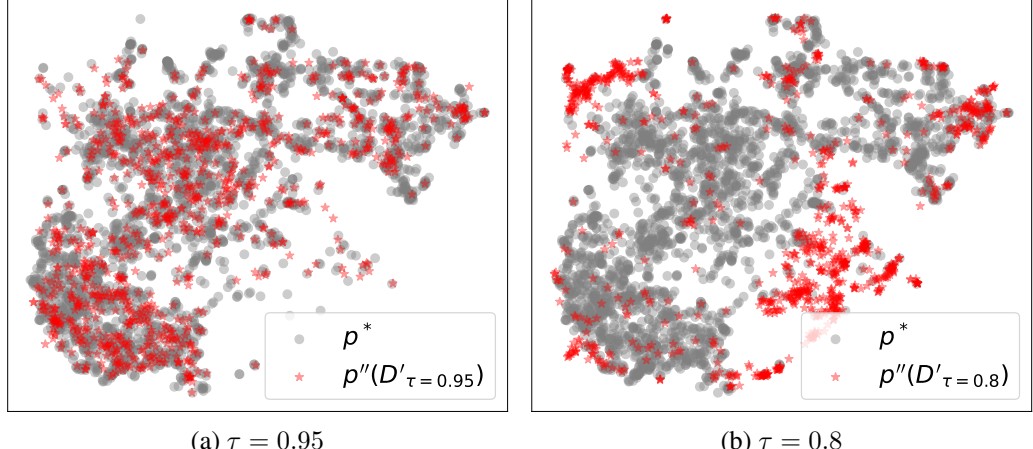

(a) $\tau = 0.95$              (b) $\tau = 0.8$

Figure 5: 2D UMAP projections of sentence embeddings generated by the ground-truth model $p^*$ (gray) and student models $p''(D'_\tau)$ (red) trained via distillation with varying teacher sampling temperatures $\tau$. As $\tau$ increases, the student's output distribution becomes more dispersed in the embedding space, covering a broader portion of the support of $p^*$. Conversely, lower $\tau$ values result in tighter clustering around specific regions, reflecting the student's emphasis on high-likelihood modes guided by the teacher's selectivity.

## 4.4 Embedding space visualization

To further examine how distillation affects the generative behavior of student models, we visualize the distribution of generated samples from $p^*$ and $p''(D'_\tau)$ in a shared embedding space using Nomic Embed v2 [18]. While the true distribution $p^*$ in language space cannot be directly plotted, sentence embeddings provide a meaningful proxy for analyzing geometric properties of generated text. Fig. 5 presents 2D projections (via UMAP [16]) of sampled sentences from two sources: the ground-truth model $p^*$ (gray) and student models trained via distillation from the teacher using different sampling temperatures (red). Each subplot corresponds to a different value of $\tau$ used during teacher sampling.

For $\tau = 0.95$, the student's generated samples broadly span the same region as $p^*$ in the embedding space, indicating relatively high recall and modest precision. In contrast, for $\tau = 0.8$, the student's samples concentrate in a narrower subregion of the $p^*$ embedding space. This tighter clustering reflects the student's increased focus on high-probability modes emphasized by the more selective teacher, consistent with higher precision and reduced recall.

These results confirm that the distributional narrowing induced by distillation—previously observed in the Gaussian mixture framework and through quantitative precision-recall metrics—also manifests geometrically in large language models. While embedding projections provide only a coarse approximation of semantic structure, they visually reinforce our interpretation of distillation as inducing controlled concentration in the student's output distribution.

We provide a comprehensive description of all experimental settings in the Appendix.

# 5 Concluding Remarks

**A principled explanation for knowledge distillation in generative modeling.** Despite its widespread adoption in generative models, knowledge distillation remains largely treated as a heuristic: a black-box technique assumed to work without a concrete understanding of how or why. In this paper, we presented what we believe to be the first principled analysis of how distillation affects generative behavior, both in theory and in practice. Starting from a controlled mixture of Gaussians setup, we demonstrated that distillation naturally induces a precision-recall trade-off governed by the entropy of the teacher distribution. Crucially, this theoretical behavior was not only consistent across simulations but also emerged clearly in experiments with autoregressive large language models.

**A minimal yet important working explanation.** By systematically varying the selectivity of the teacher and quantifying the resulting student behaviors, we provided a minimal working explanation of what knowledge distillation actually does in the generative context: it reshapes the student's distribution to concentrate on high-probability regions prioritized by the teacher. This effect is especially pronounced when the teacher exhibits low entropy (*e.g.*, via temperature scaling), leading to sharper but less diverse outputs. When sample quality is preferred over full coverage, as is common in instruction tuning, reasoning tasks, and downstream tasks, this trade-off becomes not just acceptable, but desirable. Our framework fills a critical gap in the literature by offering a geometric, distribution-level interpretation of distillation applicable to both toy and large-scale generative models.

**Limitations and future directions.** While our analysis offers a foundational step toward demystifying knowledge distillation, several limitations remain. Most notably, our LLM experiments focus on the pretraining phase, distilling from synthetic teacher data generated via next-token prediction. However, in practice, distillation is also widely applied during post-training stages, such as instruction tuning, alignment, or preference modeling. Future work should examine whether the same precision-recall trade-off persists in these settings, and how it interacts with additional fine-tuning signals. We anticipate that the underlying dynamics observed in this study will remain valid, yet they merit further validation in applied settings.

**Societal impact.** Our findings suggest that distillation can improve generation efficiency and reduce deployment cost, contributing to improved accessibility and reduced energy consumption. However, compressing generative capabilities into smaller models may also lower barriers to misuse (*e.g.*, spam or disinformation), warranting careful access control and monitoring during deployment.

In summary, we view this work as a step toward grounding the use of distillation for generative modeling in theoretical and empirical understanding. By identifying and validating a minimal working explanation, we hope to shift distillation from a rule of thumb to a principled design tool—one that can be better tuned, adapted, and relied upon in the development of future generative models.

# Acknowledgement

This work was supported by the Institute of Information & Communications Technology Planning & Evaluation (IITP) with a grant funded by the Ministry of Science and ICT (MSIT) of the Republic of Korea in connection with the Global AI Frontier Lab International Collaborative Research. This work was also supported by the Samsung Advanced Institute of Technology (under the project Next Generation Deep Learning: From Pattern Recognition to AI) and the National Science Foundation (under NSF Award 1922658). This work was supported in part through the NYU IT High Performance Computing resources, services, and staff expertise.

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

# A Appendix for Experiment Section

## A.1 Pretraining setup

We pretrain two models based on the SmolLM2 architecture to serve as the teacher and student distributions: a 360M-parameter model for the teacher distribution $p'(D)$ and a 135M-parameter model for the student distributions $p''(D'_\tau)$. Both models are trained from scratch using the same hyperparameter configuration to ensure fair and consistent evaluation.

$p'(D)$ is trained for 5 epochs using 4 NVIDIA V100 GPUs, with `DeepSpeed` [20] ZeRO Stage 2 optimization and FP16 mixed-precision enabled. The global batch size is 256, achieved using a per-device micro-batch size of 32 and a gradient accumulation step size of 2. We use the AdamW optimizer [15] with a learning rate of $5 \times 10^{-4}$, $(\beta_1, \beta_2) = (0.9, 0.95)$, and no weight decay. We apply a WSD learning rate schedule [28]: the first 1% of training steps are reserved for linear warmup, followed by a stable learning rate phase, and a final linear decay over the last 20% of steps. $p''(D'_\tau)$ is trained with the same settings, except it is only trained for one epoch. All models are initialized from the HuggingFace checkpoints: `HuggingFaceTB/SmolLM2-{size}M`, and their weights are fully reinitialized prior to training.

The training corpus consists of autoregressive language modeling prompts in JSON format. Each sample includes a single `response` string, which is tokenized with a maximum sequence length of 512 and padded to full length. We sample 100 files containing 100,000 examples each, resulting in a total of 10 million training sentences per model. Specifically, $p'(D)$ is trained on 10M samples generated by a pretrained 1.7B model with the temperature $\tau = 1$ and `top-k` set to the full vocabulary size, while each $p''(D'_\tau)$ is trained on 10M samples generated by $p'(D)$ with temperature-controlled decoding for $\tau \in \{0.8, 0.875, 0.95, 1.0\}$.

Model selection is based on perplexity evaluated on a held-out validation set of 100,000 samples ($D_{\text{val}}$). We set the temperature $\tau = 1$ and `top-k` set to the full vocabulary size when we generate validation datasets. The checkpoint with the lowest perplexity is used for evaluation. The selected $p'(D)$ is used to generate synthetic training data for distillation.

## A.2 UMAP-Based visualization setup

To analyze the distributional characteristics of the learned models, we project sentence embeddings into a 2D space using UMAP [16]. For each model distribution (*e.g.*, $p^*$, $p''(D'_\tau)$), we randomly sample 4,000 responses from $p^*$ and 1,000 responses from each $p''(D'_\tau)$, and compute their embeddings using the `all-mpnet-base-v2` model provided by Nomic Embed v2 [18]. The UMAP projection is configured with `n_neighbors=5`, `min_dist=0.001`, and `metric='cosine'`.

# B Additional Discussion

## B.1 How the precision-recall trade-off manifests across different downstream tasks

Our framework, which identifies knowledge distillation (KD) as inducing a precision-recall trade-off, can be extended to provide a principled explanation for how distilled models behave across a spectrum of distinct downstream tasks, such as summarization, reasoning, or Chain-of-Thought (CoT) generation.

We can conceptualize the modes in our generative simulation (Fig. 1a) as analogies for the data distributions of these distinct capabilities. In this view, each mode represents a specific skill or task. A large, generalist teacher model must possess broad capabilities, requiring it to cover this entire multi-modal landscape. This corresponds to achieving high recall over the complete set of tasks.

Our work demonstrates that the distillation process, particularly with a selective, lower-entropy teacher (as modulated by $\beta$ in our simulation or a low temperature $\tau$ in language models), guides the student to concentrate its probability mass on a specific subset of these modes (as empirically shown in Fig. 2c and Fig. 5). This results in a student model that achieves high precision: it develops a strong, focused competence on the targeted tasks, often matching or even exceeding the teacher's performance on those specific skills.

This gain in precision, however, occurs concurrently with a reduction in recall. The student model may lose its generalist abilities or perform poorly on the non-targeted tasks (modes) that were de-emphasized during distillation. This provides a distribution-level explanation for a widely observed empirical phenomenon: distilled student models frequently exhibit a loss of general capability while demonstrating exceptional performance on the specific tasks for which they were optimized.

To offer a concrete example, consider distilling a model for CoT reasoning. Our framework suggests that the resulting student might become highly proficient at generating a particular *style* or *format* of reasoning chain (high precision), especially if that style was dominant in the teacher's selective demonstrations. However, this student might simultaneously lose the ability to generate more diverse, exploratory, or alternative reasoning paths (low recall) that the original generalist teacher was capable of producing.

