# OpenReview forum: "Why Knowledge Distillation Works in Generative Models: A Minimal Working Explanation"
_NeurIPS.cc/2025/Conference — NeurIPS 2025 poster_

### Official Review · Reviewer_5x9X · 2025-07-01

**Clarity:** 3
**Significance:** 3
**Originality:** 3
**Rating:** 5
**Confidence:** 4

**Summary:**

The subject of the paper is Knowledge Distillation (KD), which is a fundamental machine learning technique that transfers knowledge from a large "teacher" model to a smaller "student" model. Specifically, the authors  focus on how KD works in Generative Models, and attempts to propose a minimal working explanation, as  the specific mechanisms by which KD enhances generative performance are not fully understood. Indeed, while KD explanations for classification models exist (e.g., representation alignment, label smoothing), they do not directly apply to autoregressive generative models.

The main contribution of the paper is to provide theoretical and experimental evidence that KD in generative models reshapes the student's learned distribution, inducing a precision-recall trade-off that emerges from teacher entropy.

In terms of theoretical evidence, the authors employ a Gaussian mixture setup to show that distillation shifts the student’s focus towards a selective subset of the data distribution emphasized by the teacher, leading to improved precision at the cost of recall. Then, they validate their theoretical insight in a  language model setting (using SmoLM2 1.7B as a reference, a 360M teacher model, and a 135M student model), where they show that increased teacher selectivity leads to higher precision and reduced recall. They authors claim that this suggests KD helps smaller generative models focus on high-density regions of the output space, leading to sharper and more fluent generations, which is particularly desirable in applications where sample quality is prioritized over full coverage, such as instruction tuning.

**Questions:**

No question.

**Ethical Concerns:**

["NO or VERY MINOR ethics concerns only"]

**Final Justification:**

Having read the authors' rebuttals, I still feel this is a paper worth accepting.

**Limitations:**

See weaknesses above.

**Paper Formatting Concerns:**

No concerns.

**Quality:**

3

**Strengths And Weaknesses:**

Strengths: Well-written paper that provides a novel and interesting perspective on the workings of Knowledge Distillation. I found the arguments and experiments presented convincing enough.

Weaknesses: While the theoretical and experimental evidence presented is compelling, the paper does not offer a fully rigorous explanation for the efficacy of Knowledge Distillation. Instead, it provides valuable insights and plausible explanations.

---

> ### Author Rebuttal · Authors · 2025-07-31
>
> We would like to extend our sincerest thanks to the reviewer for their very positive and encouraging feedback. We are delighted that they found our paper to be "well-written," our perspective "novel and interesting," and our arguments and experiments "convincing enough." We are grateful for their strong support of our work.
>
> We also appreciate the reviewer's thoughtful comment on the scope of our explanation, and we would like to briefly address it.
>
> ### **On the Scope of the Explanation (Weakness)**
>
> We thank the reviewer for their insightful comment: "While the theoretical and experimental evidence presented is compelling, the paper does not offer a fully rigorous explanation for the efficacy of Knowledge Distillation. Instead, it provides valuable insights and plausible explanations."
>
> We agree entirely with this characterization. Our primary goal for this work was precisely to provide what the reviewer aptly describes: "valuable insights and plausible explanations" in the form of a **"minimal working explanation"**. We aimed to establish the *first* foundational, theoretical explanation for *why* knowledge distillation is so effective in generative models, supported by rigorous empirical evidence.
>
> We believe that establishing this fundamental understanding is a crucial prerequisite before a "fully rigorous" and all-encompassing theory can be developed. For instance, we agree that this foundational framework can now serve as a crucial base to explore other important phenomena, such as explaining the success of KD in model compression techniques or inspiring the design of novel distillation methods.
>
> We see our work as building the essential groundwork upon which more complex and fully rigorous theories can be built in the future. We will revise our conclusion to better frame our contribution as this foundational first step and to more clearly articulate these exciting avenues for future research.
>
> ***
>
> Once again, we thank the reviewer for their valuable feedback and their strong support of our work.

---

> > ### Comment · Reviewer_5x9X · 2025-08-06
> > **Response to authors.**
> >
> > Thank you for your responses. I still think this is a good paper worth accepting, and I'll maintain my score. (To be clear, I agree that having a fully rigorous, end-to-end explanation, for the workings of Distillation is not necessary for accepting the paper at this stage.  I found the theoretical and experimental evidence presented by the authors convincing, and I did feel like I learned something by reading this paper.)

---

> > > ### Author Response · Authors · 2025-08-08
> > >
> > > Dear Reviewer 5x9X,
> > >
> > > Thank you for your response and for considering our rebuttal. We appreciate you taking the time to clarify your perspective and for maintaining your positive score. Your comment that you "learned something by reading this paper" was particularly encouraging to us. We are grateful for your strong support of our work.
> > >
> > > Thank you once again.
> > >
> > > Sincerely,
> > >
> > > The Authors

---

### Official Review · Reviewer_wtKg · 2025-07-03

**Clarity:** 4
**Significance:** 3
**Originality:** 4
**Rating:** 6
**Confidence:** 4

**Summary:**

This paper presents a minimal and intuitive explanation for the effectiveness of knowledge distillation (KD) in generative models. The core contribution is the identification of a fundamental trade-off between precision and recall, which is induced and controlled by the teacher model's entropy. Through a controlled simulation with Gaussian Mixture Models, the authors demonstrate that a more selective (lower-entropy) teacher guides the student model to concentrate its probability mass on high-likelihood regions of the data distribution, thereby improving sample quality (precision) at the expense of distributional coverage (recall). The paper compellingly shows that this same mechanism is at play in large-scale language models, providing a principled explanation for why KD is so effective in practice, especially for tasks where generation quality is prioritized over diversity.

**Questions:**

Would there be distinctive characteristics of KD for different tasks, such as summarization, reasoning, and classification? For example, large reasoning models generate tokens to contextualize the thought process, and it is unclear how knowledge distillation can fit into this chain-of-thought setting. Any insights on how such task-dependent characteristics would affect the role of KD would be very insightful for the community.

**Ethical Concerns:**

["NO or VERY MINOR ethics concerns only"]

**Final Justification:**

After reviewing all the comments and rebuttals, I maintain my original positive rating. IMHO, this work differs from prior studies [1,2,3] in an important way: while existing works focus on proposing new techniques supported by specific observations or insights, this work provides a novel perspective for understanding the fundamentals of knowledge distillation (KD). I believe this new viewpoint will help researchers better understand the success and failure modes of KD techniques, particularly when applied to diverse task-specific scenarios.

The additional discussion the authors provided in the rebuttal regarding "how the precision-recall trade-off might manifest across different downstream tasks" would be valuable for readers if formally incorporated into the paper's discussion section or appendix.

**Limitations:**

Yes

**Quality:**

4

**Strengths And Weaknesses:**

Strengths
- The use of a Gaussian Mixture Model as a "minimal working explanation" is a brilliant methodological choice. It allows the authors to deconstruct the complex dynamics of KD into an analytically tractable problem, providing a clear and intuitive foundation for their claims. The subsequent validation on large-scale SmolLM2 models demonstrates rigorous research practice, successfully bridging the gap from a simplified theoretical model to a complex, real-world application.

- The concepts of precision and recall are defined clearly and used effectively throughout the paper to quantify the trade-off. The visualizations (Figures 2 and 4) are particularly effective, offering a strong, intuitive visual confirmation of the quantitative results and reinforcing the paper's core message.

- This work provides what is arguably the first principled, distribution-level explanation for why KD works in this domain. By framing KD's role as reshaping the student's output distribution, the paper moves beyond heuristic explanations and offers a novel, geometric interpretation.

- The most significant contribution is providing a theoretical rationale for a widely observed empirical phenomenon. The authors clearly articulate why KD is so beneficial for tasks like instruction tuning, where the goal is to generate high-quality, coherent outputs (high precision) rather than exploring the full diversity of the language space (high recall). This insight is highly valuable to practitioners and researchers alike, offering a new lens through which to view and design distillation pipelines.

Weaknesses / Areas for Improvement
- The paper's analysis focuses on how KD helps a smaller student model mimic the behavior of a larger teacher. However, KD is also a cornerstone of model compression techniques, such as quantization (i.e., reducing the bit precision of model weights). The current framework does not explicitly address how this precision-recall trade-off might explain the success of KD in such scenarios.

- While the paper provides a powerful explanation, it stops short of proposing new methods based on its findings. The insights gained could potentially be leveraged to design more sophisticated and effective distillation strategies.

---

> ### Author Rebuttal · Authors · 2025-07-31
>
> We are sincerely grateful to the reviewer for their exceptionally positive and encouraging feedback. We are thrilled that they found our work to be of "excellent" quality, clarity, and originality, and we deeply appreciate them highlighting our contributions, such as the use of GMMs as a "brilliant methodological choice" and our work providing the "first principled, distribution-level explanation for why KD works in this domain".
>
> We also thank the reviewer for their thoughtful suggestions on areas for improvement and their insightful question. We believe these points will help us enrich the discussion in our final manuscript.
>
> ### **On Weaknesses and Future Directions**
>
> We thank the reviewer for their excellent points regarding the broader applications of our framework, such as explaining KD's success in model compression (e.g., quantization) and its potential for inspiring new distillation methods.
>
> Our primary goal in this paper was to establish the **first foundational, theoretical explanation for *why* knowledge distillation is so effective in generative models**, supported by rigorous empirical evidence. We believe that establishing this core understanding is a crucial prerequisite for the very directions the reviewer suggests.
>
> Our **ground-truth → teacher → student** framework provides a new analytical lens for the community. By demonstrating that introducing an intermediate teacher stage can fundamentally alter the learning outcome even with the same forward KL loss, we open up new avenues for investigation. We strongly agree that this framework could serve as a theoretical and empirical base to:
> 1.  Analyze why KD is successful in model compression techniques like quantization.
> 2.  Inspire the design of novel, more sophisticated distillation strategies or loss functions.
>
> We see these as very promising and exciting directions for future research. We will update the "Limitations and Future Directions" section of our paper to explicitly mention these possibilities and frame our work as a foundational step toward them.
>
> ### **Response to Questions**
>
> This is a very insightful question regarding how the precision-recall trade-off might manifest across different downstream tasks, such as summarization, reasoning, and Chain-of-Thought (CoT) generation. Our framework provides a direct way to reason about this.
>
> The modes in our Gaussian mixture model (**Figure 1a** of the manuscript) can be viewed as analogies for different capabilities or downstream tasks. In this view, each mode represents the data distribution for a specific skill (e.g., one mode for summarization, another for reasoning). A generalist model would need to cover all modes (high recall).
>
> Our work demonstrates that distillation, especially with a selective teacher, guides the student to focus intensely on a subset of these modes (as shown in **Figure 2c** of the manuscript). This leads to a student model that achieves very high performance on the targeted tasks (high precision) but may lose some of its general capability across the other, non-targeted tasks (low recall).
>
> This explains a widely observed empirical phenomenon: distilled student models often lose some of their generalist abilities but can match or even exceed teacher performance on the specific tasks they were distilled for. Our framework provides a principled, distribution-level explanation for this behavior. For a task like CoT, this implies that distillation could create a student that is highly proficient at generating a specific *style* of reasoning chain (high precision), but it might lose the ability to generate more diverse or exploratory reasoning paths (low recall).
>
> ***
>
> Once again, we are deeply grateful for the reviewer's positive assessment. Your thoughtful feedback will help us improve the final version of the paper.

---

> ### Author Response · Authors · 2025-08-06
>
> Dear Reviewer wtKg,
>
> This is a gentle reminder that the discussion period will close in approximately 36 hours. We would be very grateful if you could take a moment to review our rebuttal and provide any final feedback.
>
> Thank you again for your time and valuable contributions to our work.
>
> Sincerely,
>
> The Authors

---

> ### Comment · Reviewer_wtKg · 2025-08-08
>
> After reviewing all the comments and rebuttals, I maintain my original positive rating. IMHO, this work differs from prior studies [1,2,3] in an important way: while existing works focus on proposing new techniques supported by specific observations or insights, this work provides a novel perspective for understanding the fundamentals of knowledge distillation (KD). I believe this new viewpoint will help researchers better understand the success and failure modes of KD techniques, particularly when applied to diverse task-specific scenarios.
>
> The additional discussion the authors provided in the rebuttal regarding "how the precision-recall trade-off might manifest across different downstream tasks" would be valuable for readers if formally incorporated into the paper's discussion section or appendix.

---

> > ### Author Response · Authors · 2025-08-08
> >
> > Dear Reviewer wtKg,
> >
> > Thank you for taking the time to review our rebuttal and provide additional thoughtful feedback. We are sincerely grateful for your strong advocacy and for articulating a clear case for our work's novelty.
> >
> > We agree that your suggestion is excellent. We will be sure to incorporate the discussion on how the precision-recall trade-off manifests across different downstream tasks into the final version of the manuscript.
> >
> > Thank you again for your valuable contributions.
> >
> > Sincerely,
> >
> > The Authors

---

### Official Review · Reviewer_sQ8t · 2025-07-05

**Clarity:** 2
**Significance:** 2
**Originality:** 1
**Rating:** 2
**Confidence:** 4

**Summary:**

This paper provides a minimal working explanation for why knowledge distillation (KD) is effective in generative models. It argues that KD induces a fundamental trade-off between precision and recall in the student model. Using both a controlled Gaussian mixture simulation and large-scale language model experiments , the work demonstrates that a more selective (lower-entropy) teacher distribution forces the student to concentrate probability mass on high-likelihood regions. This enhances sample quality (precision) at the expense of distributional coverage (recall). This dynamic is particularly desirable in applications where sample quality outweighs diversity, such as instruction tuning.

**Questions:**

see weaknesses.

**Ethical Concerns:**

["NO or VERY MINOR ethics concerns only"]

**Limitations:**

yes

**Quality:**

2

**Strengths And Weaknesses:**

Strength:

This work bridges a simple theoretical model with a large-scale, practical application to show the precision-recall trade-off in knowledge distillation.
This work demonstrates that precision-recall trade-off is often a desirable feature, not a bug, is highly valuable. It explains why KD is so effective in applications like instruction tuning
Weakness:

The main insight from the paper "the precision-recall trade-off in KD" is already demonstrated in many preivious works studying KD in generative models [1,2,3]. For example [1] states that KD introduces a mode-seeking behavior, which forces the student model to search for the largest modes in the teacher model and is critical in practical scenarios that require truthfulness and reliability. I do not see other more insights from this paper beyond these previous works.

There is a confusing misalignment between the theory in this paper and previous literatures. Equation (2) represents the forward KLD between the teacher model and the student model, which, as shown in [1] and [3], will introduce a mode-averaging behavior and harm precision. Could the authors explain the misalignment?

(2 continued) There is also a misalignment between the theory and the simulation results in this paper. From Equation (4), and the simulation results, $E{p^(x)}\left[\log p''(x)\right]$ will decrease after distillation, which means $$ KL[p^(x)||p''(x)]=-E{p^(x)}\left[\log p''(x)\right]+\text{const} $$ will increase. When $K'=K$, the teacher model will exactly fit the real distribution $p^$, making $KL[p'(x)||p''(x)]=KL[p^*(x)||p''(x)]$. In this way, the KD loss in Equation (2) will also increase. This introduces the contradiction because KD aimes at minimizing Equation (2). Could the authors give an explanation on this?

The theory seems problematic. The analysis after Equation (2) shows an upper bound showing the mode-seeking behavior for $KL[p'(x)||p''(x)]$. However, this does not suffciently guarantee that $KL[p'(x)||p''(x)]$ will also have the mode-seeking behavior. There may exisit better solutions for $\min KL[p'(x)||p''(x)]$ with other behaviors.

Missing important references in KD for generative models [1,2,3].

References:

[1] MiniLLM: Knowledge Distillation of Large Language Models.

[2] On-Policy Distillation of Language Models: Learning from Self-Generated Mistakes.

[3] f-Divergence Minimization for Sequence-Level Knowledge Distillation

---

> ### Author Rebuttal · Authors · 2025-07-31
>
> We sincerely thank the reviewer for their time and for providing a detailed and thoughtful review. We are grateful for the positive feedback acknowledging the strengths of our work, particularly that it "bridges a simple theoretical model with a large-scale, practical application" and that our demonstration of the precision-recall trade-off as a "desirable feature, not a bug, is highly valuable".
>
> We appreciate the opportunity to address the weaknesses and questions raised. We believe these points stem from a few key misunderstandings of our paper's core contributions and methodology. We will address each concern below and have committed to significant revisions to clarify these points in the manuscript.
>
>
> ### **1. On Originality and the Precision-Recall Trade-off (Weakness 1 & 4)**
>
> We thank the reviewer for pointing out several important references[1, 2, 3]. However, we respectfully disagree with the assessment that our main insight is "already demonstrated" in these works. Our contribution is fundamentally different in both its **method** and its **analytical scope**.
>
> *   **A Novel Mechanism with Standard Forward KL:** Our core contribution is demonstrating that a mode-seeking behavior can be induced using **standard forward KL divergence**—the default for many KD setups—simply by controlling the teacher's entropy. This is a distinct and novel mechanism compared to prior work like MiniLLM [1, 3], which proposes using a special objective, **reverse KL divergence**, or a sophisticated loss function [2], specifically to achieve mode-seeking. This is crucial because it provides a principled explanation for the effectiveness of the most common form of KD, rather than requiring specialized objectives. Our work explains the behavior of the tool that practitioners are already using.
>
> *   **A More Comprehensive Analytical Framework:** Prior works, including those cited, primarily analyze the relationship between the teacher and the student. Our paper introduces a more holistic three-stage analysis: **ground truth → teacher → student**. We are the first to theoretically and empirically show how a student model, trained with standard forward KD, exhibits mode-seeking behavior *relative to the original ground-truth distribution*. This broader perspective allows us to provide "a minimal working explanation" for *why* KD is effective in improving sample quality in generative models, a question that previous teacher-student analyses do not fully address.
>
> In summary, while the goal of achieving precision is shared, our work offers a novel and complementary perspective to methods like MiniLLM. Where they modify the objective function, we demonstrate how to achieve similar desirable outcomes by controlling the teacher's properties within the standard KD framework. Our analysis of the three-stage framework provides a more general explanation for the effectiveness of KD. We thank the reviewer again for these valuable references and will add a new subsection to our Related Work section to discuss these papers and better contextualize our unique contributions.
>
> ### **2. On the Alleged Misalignment of Forward KL (Weakness 2)**
>
> We agree with the reviewer's correct assertion that forward KL divergence ($KL(P||Q)$) generally exhibits "mass-covering" or "mean-seeking" behavior. In fact, our paper explicitly demonstrates this very phenomenon. As shown in **Figure 1** of the manuscript, when a simple student model (unimodal Gaussian) is trained directly on a multi-modal ground-truth distribution, it indeed averages the modes, resulting in a low-quality fit where probability mass is placed in low-density regions of the true distribution.
>
> However, our paper's analysis does not stop at this simple two-way relationship. Our core contribution lies in analyzing the full **ground-truth → teacher → student** pipeline. As shown in **Figure 2**  of the manuscript, our process is as follows:
> 1.  A teacher model with capacity $K' < K$ is trained on the ground-truth data, learning a subset of the modes (**Fig. 2a**).
> 2.  This teacher is made more selective (low-entropy) via our $\beta$ parameter.
> 3.  A student model with capacity $K'' < K'$ is then trained on this selective teacher using standard forward KL. The result is a student that focuses its mass on the single, dominant mode presented by the teacher (**Fig. 2c**).
>
> Crucially, the final student model exhibits high precision and low recall *with respect to the original ground-truth distribution*. Therefore, our work shows how the standard forward KL objective, when used within this multi-stage framework, results in a desirable mode-seeking behavior in the final student model. There is no misalignment; rather, we provide a more complete picture that explains how this emergent behavior arises. This is precisely the "minimal working explanation" we aimed to provide—showing how a seemingly simple setup can explain a complex and valuable phenomenon in generative model distillation.
>
> ### **3. On the Alleged Contradiction in Simulation Results (Weakness 2 continued)**
>
> The reviewer correctly observes from our results that as the KD training loss $KL(p'\_{teacher}||p''\_{student})$ is minimized, the Recall metric (related to $KL(p*\_{ground-truth}||p''\_{student})$) worsens. The reviewer interprets this as a contradiction.
>
> We respectfully clarify that this is **not a contradiction, but the central finding of our paper**. We explicitly define and demonstrate a trade-off where the student model intentionally sacrifices coverage of the full ground-truth distribution ($p^*$) to achieve higher precision by faithfully mimicking the selective, low-entropy teacher ($p'$).
>
> Furthermore, the reviewer's hypothetical scenario for this contradiction is based on an invalid premise. The reviewer states: "When $K'=K$, the teacher model will exactly fit the real distribution $p^*$." This is incorrect for two reasons:
> 1. **Practical Impossibility**: In any real-world scenario, the number of modes in the ground-truth distribution, $K$, is unknown. Therefore, setting the teacher's capacity $K'$ to be equal to $K$ is not practically possible.
> 2. **Inherent Learning Noise**: As we explicitly state in our paper, even in the ideal case of $K′=K$, the teacher parameters are recovered only "up to some noise due to the finite number of samples within $D$" (see Line 113 of the manuscript).  This inherent noise from statistical learning ensures that the teacher $p^′$ would never be a perfect replica of the ground-truth $p^∗$, even with perfectly matched capacity.
> 3.  **Methodological Design**: Most importantly, even if we could assume $K'=K$, our method's application of the parameter $\beta > 1$ ensures that the teacher distribution $p'(x;\beta)$ is a lower-entropy modification and thus does not exactly fit the real distribution $p^*(x)$.
>
> Therefore, the situation described by the reviewer cannot occur under our framework. Additionally, we believe that the core of this misunderstanding appears to be a conflation of the training objective ($KL(p'\_{teacher}||p''\_{student})$) with an evaluation metric (Recall, which is related to $KL(p*\_{ground-truth}||p''\_{student})$). Our method successfully trains the student to minimize its divergence from the selective teacher, which is the explicit goal of Equation (2). The fact that this simultaneously increases the student's divergence from the full ground-truth distribution is not a flaw in the optimization; it is the very definition of the precision-recall trade-off that our paper identifies and explains.
>
> ### **4. On the "Problematic Theory" (Weakness 3)**
>
> We appreciate the reviewer's close reading of our theoretical analysis. We would like to clarify two key points regarding the analysis following Equation (2), which we believe will resolve the reviewer's concerns.
>
> 1. **Correction on the Bound Type**: First, we would like to clarify a small but important technical point. The reviewer states that our analysis shows an "upper bound". However, because the logarithm is a concave function, the application of Jensen's inequality to the integral term results in a lower bound on that term, which our optimization seeks to maximize. This is equivalent to minimizing an upper bound on the KL divergence loss itself.
> 2. **Clarification on the Analysis's Purpose**: Second, and more importantly, this addresses the reviewer's core concern about the analysis's purpose. The purpose of this analysis was not to serve as a formal proof or to "sufficiently guarantee" the mode-seeking behavior. As we state in the paper, its goal is to "better understand this objective". The decomposition is intended purely as an illustration to provide intuition for how the optimization encourages the student to align with the teacher's high-density modes.
>
> The mode-seeking behavior itself does not arise from this Jensen's inequality analysis. Instead, it is a direct consequence of our methodological setup: we intentionally construct a low-entropy, selective teacher distribution ($p^′$). The analysis via the bound simply helps to explain the mechanics of how the student model's optimization is guided by this pre-conditioned, selective teacher.
> Therefore, the theory is not problematic; the analysis serves its intended illustrative purpose, and the mode-seeking behavior is a result of the problem formulation itself. We will revise the text to make the illustrative purpose of this section even clearer.
>
> ***
>
> We hope these clarifications address the reviewer's concerns. We are confident that by incorporating the suggested references and clarifying our methodology and contributions as outlined above, the revised manuscript will be significantly stronger. We thank the reviewer again for their valuable feedback.

---

> ### Author Response · Authors · 2025-08-06
>
> Dear Reviewer sQ8t,
>
> This is a gentle reminder that the discussion period will close in approximately 36 hours. We would be very grateful if you could take a moment to review our rebuttal and provide any final feedback.
>
> Thank you again for your time and valuable contributions to our work.
>
> Sincerely,
>
> The Authors

---

> ### Author Response · Authors · 2025-08-08
>
> Dear Reviewer sQ8t,
>
> We hope this message finds you well. We noticed the author-reviewer discussion period has been extended, and we would be very grateful if you could take a moment to review our rebuttal and the ongoing discussion.
>
> We also wanted to gently note that other reviewers have since shared their thoughts after reading our response. We believe their comments, alongside our rebuttal, might help clarify our contributions. We would sincerely appreciate any final feedback you may have.
>
> Thank you again for your time and your valuable role in this process.
>
> Sincerely,
>
> The Authors

---

### Note · Authors · 2025-08-14

Dear Area Chair and Reviewers,

We would like to express our sincere gratitude for the comprehensive review process. We are particularly grateful to **Reviewers wtKg (6/6) and 5x9X (5/6)** for their strong, articulated support and for engaging in a constructive discussion that has helped clarify the value of our work.

The primary unresolved issue remains the 'Reject' recommendation from **Reviewer sQ8t**. We provided a detailed rebuttal to address what we believe are fundamental misunderstandings of our work. Crucially, **the reviewer's main concern regarding originality was also directly refuted by Reviewer wtKg in the open discussion**, who explained how our work is fundamentally different from the prior works[1,2,3].

Despite this, and our multiple reminders, **Reviewer sQ8t did not engage in the discussion period**. This lack of engagement prevented any opportunity to resolve the technical misunderstandings. We also observe that the reviewer's "Mandatory Acknowledgement" **after ending the author-reviewer discussion**, indicating they "engaged in discussions and responded to authors", does not seem to align with the discussion history on OpenReview.

In summary, this paper provides the first principled explanation for why standard knowledge distillation (using forward KL) works well in generative models, in terms of both theoretical and experimental aspects. We are confident in the supportive assessments from the majority of our expert reviewers.

We trust the committee will consider the full context of the review process, including the level of reviewer engagement, in its final decision.

Sincerely,

The Authors

---

### Decision · Program_Chairs · 2025-09-17

**Decision:**

Accept (poster)

**Comment:**

The authors use a GMM simulation and large-scale language model experiments to show that standard knowledge distillation induces a trade-off: it helps the student model achieve high precision by focusing on high-probability regions of the teacher's distribution, but at the cost of recall. The authors argue this trade-off is often desirable, especially for tasks like instruction tuning where a single, high-quality answer is preferred over a diverse set of possible outputs.

Strengths
- The paper provides a principled, distribution-level explanation for a widely observed phenomenon in knowledge distillation. This is a significant contribution that moves beyond heuristic explanations and offers a new framework for understanding and designing distillation pipelines. The use of a simple GMM model to provide an intuitive "minimal working explanation" is particularly effective and praised by multiple reviewers.
- The authors rigorously validate their theoretical findings on a large-scale language model family (SmolLM2), successfully bridging the gap between a simplified theoretical model and a practical application. The experiments and visualizations are clear and convincing.
- The paper is well-written and easy to follow, clearly defining key concepts like precision and recall and effectively communicating its core insights.

Weaknesses
- The central insight, the precision-recall trade-off, is not new and has been discussed in prior works. While the authors' rebuttal clarifies that the paper's novelty lies in achieving this behavior using standard forward KL divergence without specialized loss functions, this point remains a source of contention.
- Reviewer raises questions about the theoretical justification and a potential contradiction between the model's objective (minimizing KL divergence) and its behavior (sacrificing recall). While the authors provide a detailed response clarifying that the recall sacrifice is the point of the trade-off, not a flaw, the reviewer remains unconvincingly and continues to question the theoretical claims.
- Some reviewers note that the paper could be expanded to explore KD's role in other applications like model compression (e.g., quantization) or its effects on different tasks like reasoning with CoT prompting. The authors acknowledge these are valuable directions for future work.

The AC recommends a borderline accept decision, noting that the paper has strong conceptual and empirical merit but is weakened by the persistent, unaddressed theoretical concerns from one reviewer. The paper's contribution is valuable and likely to be impactful for practitioners and researchers, but its theoretical foundations could be more robustly justified.